# mTOR Inhibition Impairs the Activation and Function of Belatacept-Resistant CD4^+^CD57^+^ T Cells In Vivo and In Vitro

**DOI:** 10.3390/pharmaceutics15041299

**Published:** 2023-04-20

**Authors:** Florence Herr, Manon Dekeyser, Jerome Le Pavec, Christophe Desterke, Andrada-Silvana Chiron, Karen Bargiel, Olaf Mercier, Amelia Vernochet, Elie Fadel, Antoine Durrbach

**Affiliations:** 1Institut Gustave Roussy, Inserm, Immunologie Intégrative des Tumeurs et Immunothérapie des Cancers, Université Paris-Saclay, 94805 Villejuif, France; 2Hôpital Henri Mondor, Service de Néphrologie, Assistance Publique-Hôpitaux de Paris, 94010 Creteil, France; 3Inserm, Hypertension Pulmonaire: Physiopathologie et Innovation Thérapeutique, Université Paris-Saclay, 92350 Le Plessis Robinson, France; 4Centre Hospitalier Marie Lannelongue, 92350 Le Plessis Robinson, France; 5Inserm, Modèles de Cellules Souches Malignes et Thérapeutiques, Université Paris-Saclay, 94805 Villejuif, France; 6Unité des Technologies Chimiques et Biologiques pour la Santé, CNRS, INSERM, UTCBS, Université de Paris, 75006 Paris, France; 7Clinical Immunology Laboratory, Groupe Hospitalier Universitaire Paris-Sud, Hôpital Kremlin-Bicêtre, Assistance Publique-Hôpitaux de Paris, 94270 Le Kremlin-Bicetre, France

**Keywords:** solid-organ transplantation, checkpoint inhibitors, mTOR inhibitors, belatacept-resistant T cells

## Abstract

Calcineurin inhibitors have improved graft survival in solid-organ transplantation but their use is limited by toxicity, requiring a switch to another immunosuppressor in some cases. Belatacept is one option that has been shown to improve graft and patient survival despite being associated with a higher risk of acute cellular rejection. This risk of acute cellular rejection is correlated with the presence of belatacept-resistant T cells. We performed a transcriptomic analysis of in vitro-activated cells to identify pathways affected by belatacept in belatacept-sensitive cells (CD4^+^CD57^−^) but not in belatacept-resistant CD4^+^CD57^+^ T cells. mTOR was significantly downregulated in belatacept-sensitive but not belatacept-resistant T cells. The inhibition of mTOR strongly decreases the activation and cytotoxicity of CD4^+^CD57^+^ cells. In humans, the use of a combination of mTOR inhibitor and belatacept prevents graft rejection and decreases the expression of activation markers on CD4 and CD8 T cells. mTOR inhibition decreases the functioning of belatacept-resistant CD4^+^CD57^+^ T cells in vitro and in vivo. It could potentially be used in association with belatacept to prevent acute cellular rejection in cases of calcineurin intolerance.

## 1. Introduction

Solid-organ transplantation (SOT) has become the best strategy for overcoming chronic end-stage organ failure for heart, lung, liver and renal disease. Progress in immunosuppression has improved graft function while decreasing rates of acute cellular rejection. Calcineurin inhibitors (CNIs), which were introduced in the 1980s, have played an important role, making it possible to control T-cell activation, and leading to a major decrease in cellular graft rejection rates at one year and improvements in graft survival [1]. However, the benefits of controlling adaptive immunity are countered by the long-term nephrotoxicity of CNIs and by increases in cardiovascular risk. These drawbacks have led to the development of new molecules. Based on the theory of three activation signals for T cells, belatacept—a modified CTLA4-Ig molecule with a high affinity for CD80/86—was developed to inhibit T-cell activation in the absence of CNIs, by inhibiting the CD28 pathway, which provides the second signal for T-cell activation. This drug is now approved for the prevention of acute rejection in patients undergoing standard kidney transplantation or receiving a kidney from an extended-criteria donor. Belatacept has been shown to improve renal function three, five and seven years post transplantation, an important predictive marker of long-term graft survival. Patients receiving belatacept have been confirmed to have higher graft and global survival rates, and a lower cardiovascular risk than patients on cyclosporin A [2,3,4,5].

However, belatacept is associated with a high rate of acute cellular rejection (ACR), and with a higher grade of ACR, mostly due to infiltrating adaptive immune cells, despite lower rates of de novo donor-specific anti-HLA antibodies (dnDSA) [6]. Several groups have analyzed the phenotype of the T cells associated with ACR in patients treated with belatacept, to explore the higher incidence of ACR and to identify T-cell populations resistant to belatacept. Two CD4^+^ T-cell populations with a memory phenotype have been associated with the development of ACR in belatacept-treated patients, with greater expansion of the CD8^+^CD28^neg^ T-cell population at the time of ACR [7]. A higher risk of ACR was observed in patients with high levels of CD4^+^CD28^+^ effector memory T cells before transplantation [7]. Similarly, we found that a high percentage of the CD4^+^CD57^+^PD1^−^ memory T-cell population associated with ACR. CD4^+^CD57^+^PD1^−^ cells are belatacept-resistant and highly cytotoxic, and have been shown to infiltrate allogeneic kidneys during rejection [8,9,10].

Belatacept-resistant cells have been shown to proliferate in the presence of belatacept, indicating that this drug does not prevent these cells from entering the cell cycle and undergoing clonal expansion [11,12,13,14]. Progression of the cell-cycle, reparation of DNA, genetic stability and ribosomal RNA synthesis, among other critical process for cell proliferation, are regulated by the PI3K/AKT/mTORC pathway. mTOR interacts with other proteins forming mTOR complex (mTORc) 1 or 2, which have different regulation and downstream effects. Numerous compounds have been developed as an mTOR inhibitor (mTORi). The first generation of mTORi, derived from rapamycin, target the catalytic site of mTOR, inhibiting mTORc1 formation. Rapamycin and the derivatized molecule everolimus are used in transplantation for their immunosuppressive activity [13,14,15]. These molecules were initially developed to prevent ACR, but their ACR-preventing properties were not sufficiently strong for them to replace CNIs. However, pilot studies suggested that the combination of mTORi with belatacept effectively decreased the risk of developing ACR [16,17,18]. We analyzed pathways of activation not impaired by the second signal of inhibition in belatacept-resistant T cells. We show that mTOR inhibition can reduce belatacept-resistant T-cell proliferation and control T cell-mediated rejection in patients displaying belatacept-resistant rejection.

## 2. Materials and Methods

### 2.1. Cell Culture

Peripheral blood mononuclear cells (PBMCs) from healthy volunteers collected at the Etablissement Français du Sang were isolated by cytapheresis followed by Pancoll (PANBiotech, Aidenbach, Germany) density gradient centrifugation. All donors gave written informed consent. Cells were cultured in RPMI 1640 medium containing 2 mM stabilized L-glutamine (Sigma-Aldrich, St. Louis, MO, USA), 10% heat-inactivated fetal bovine serum (FBS, Capricorn Scientific, Ebsdorfergrund, Germany), 100 U/mL penicillin, and 100 µg/mL streptomycin (Sigma-Aldrich, St. Louis, MO, USA). For experiments on T cells from patients with end-stage renal disease (ESRD), we used frozen PBMCs, which were thawed at 37 °C and washed in PBS supplemented with 20% FBS before use.

### 2.2. Dendritic Cell Differentiation

PBMC-derived monocytes were isolated with the MojoSort Human CD14 Selection kit (BioLegend, San Diego, CA, USA), according to the manufacturer’s protocol. CD14+ cells (purity > 98% assessed by flow cytometry) were plated at 0.3 × 10^6^ cells/cm^2^ in complete RPMI 1640 medium supplemented with 1000 U/mL rhGM-CSF (Peprotech, London, UK) and 250 U/mL rhIL-4 (Peprotech, London, UK) and cultured for six days at 37 °C with 5% CO_2_, as previously described. Immature dendritic cells (DCs) were activated with 50 ng/mL lipopolysaccharide (LPS, from *E. coli* O127:B7, Sigma-Aldrich, St. Louis, MO, USA) in the same medium, at 37 °C, 5% CO_2_, for two additional days. On days 6 and 8, DCs were analyzed for their expression of CD14, CD209, HLA-DR, CD80 and CD83 by flow cytometry.

### 2.3. Mixed Lymphocyte Reaction (MLR)

T lymphocytes were selected from PBMCs by negative selection with the MojoSort Human CD3 T-cell isolation kit (BioLegend, USA) (enrichment >95% assessed by flow cytometry). Cells were sorted on the basis of their CD57 expression, with an FACS-Aria sorter. T cells were stained with 1 µM Violet Proliferation Dye 450 (VPD450; BD Biosciences, San Jose, CA, USA) in Dulbecco’s phosphate-buffered saline (DPBS, Sigma-Aldrich, St. Louis, MO, USA) for 10 min at 37 °C, and were then washed in complete medium. CD57^+^ VPD450^+^ cells and CD57^−^ VPD450^−^ cells were mixed together in a 1:10 ratio and were incubated with allogeneic-activated DCs (aDCs), (DC:T-cell ratio of 1:5), in RPMI 1640 complete medium, at 37 °C, 5% CO_2_. According to the experiments, belatacept (Bristol Myers Squibb, New York, NY, USA) was added at a concentration of 250 µg/mL in the complete medium of aDC 15 min prior MLR with a subsequent final concentration of 25 µg/mL and everolimus at a concentration of 5 ng/mL. The proliferation and phenotype of the T cells were analyzed by flow cytometry after 5 days of MLR.

For RNA-seq experiments, isolated CD3^+^ T cells were cultured for 6 h at a density of 2 × 10^6^ cells/mL with allogeneic aDCs in complete medium, at 37 °C, under an atmosphere containing 5% CO_2_, with a DC:T-cell ratio of 1:5. After MLR, the cells were sorted on the basis of their CD4, CD57 and PD1 expression, with an FACS-Aria sorter (BD Biosciences, San Jose, CA, USA), and were immediately washed in DPBS, pelleted by centrifugation and frozen at −80 °C.

### 2.4. Flow Cytometry

The following antibodies against surface antigens and their control isotypes were used: anti-CD3 APC-Fire770, anti-CD57 PE or AF647, anti-PD1 PE-Cy7 or BV786, anti-CD25 APC-Cy7, anti-CD8 BV510 (all from BioLegend, San Diego, CA, USA), anti-CD4 BUV395, anti-CCR7 BV605, anti-CD3 FITC, anti-CD38 PE-Cy7 or BB515, anti-CD45RO BV650, anti-CD25 APC, anti-CD8 PerCP-Cy5.5 (all from BD Biosciences, San Jose, CA, USA). The memory phenotypes of T cells were defined as naïve (CCR7^+^CD45RO^−^), central memory (CM; CCR7^+^CD45RO^+^), effector memory CD45RA^+^ (EMRA; CCR7^−^CD45RO^−^) or effector memory (EM; CCR7^−^CD45RO^+^) cells.

Briefly, PBMCs from healthy donors or dialysis patients, or MLR cells were washed twice in staining buffer (2% FBS, 0.1% sodium azide in DPBS) and stained by incubation with fluorochrome-conjugated antibodies or isotypic control antibodies for 30 min at 4 °C. The cells were then washed twice with staining buffer and analyzed.

Granzyme B analysis was performed with anti-granzyme B-FITC or -AF700 (BD Biosciences, San Jose, CA, USA) antibodies. Cells from the MLR were incubated for 4 h at 37 °C, under an atmosphere containing 5% CO_2_, with target cells (PBMCs from the corresponding aDC donor) in a 1:1 ratio, in the presence of Golgi-Stop (Monensin, BD Biosciences, San Jose, CA, USA). The cells were then stained for surface antigens, as described above, fixed with the BD Cytofix-Cytoperm-Plus kit (BDBiosciences, San Jose, CA, USA), according to the manufacturer’s protocol and stained for intracellular markers. For the analysis of CD107a expression, cells from the MLR were incubated for 4 h at 37 °C, under an atmosphere containing 5% CO_2_, with target cells (PBMCs from the corresponding aDC donor) in a 1:1 ratio, in the presence of anti-CD107a FITC antibody (BD Biosciences, San Jose, CA, USA). The cells were then stained for surface antigens, as described above. S6 phosphorylation and mTOR expression were assessed on PBMC from healthy donors. For phospho-S6 analysis cells were incubated in OKT3 (1 µg/mL, Biolegend) pre-coated wells 5 min at 37 °C, as indicated. Then, cells were fixed in 2% paraformaldehyde for 15 min at 37 °C and stained for surface antigens. For mTOR analysis cells were stained for surface antigens before fixation. Permeabilization was performed with True-Phos perm Buffer (Biolegend) and stained with anti-phospho-S6 (Ser235/236) and anti-rabbit-PE or anti-mTOR-PE (Cell Signaling Technology) antibodies, according to the manufacturer protocol.

Cells were characterized on BD LSR Fortessa flow cytometer (BD Biosciences, San Jose, CA, USA) and data were analyzed with FlowJo software V10.8.1 (Becton, Dickinson and Company, Franklin Lakes, NJ, USA).

### 2.5. RNA-seq Analysis and RNA-seq Data Processing

RNA was isolated from cell pellets with the RNeasy-Kit (Qiagen, Germantown, MD, USA), according to the manufacturer’s instructions. For details, see the Appendix A.

Raw data were demultiplexed and FASTQ files were generated for each sample with bcl2-fastq software (Illumina Inc., San Diego, CA, USA). The FASTQ data were checked with the FastQC tool. Data analysis was performed with the Qiagen-GeneGlobe bioinformatics tool. Sequences were aligned with the *GRChv38* human reference genome. Secondary differential expression analysis was performed with DESeq2, with a normalization of UMI counts (variance stabilization transformation, VST), followed by pairwise differential regulation analysis. The expression profiles of belatacept-treated or untreated CD4^+^CD57^−^, CD4^+^CD57^+^PD1^+^ and CD4^+^CD57^+^PD1^−^ cells were analyzed by gene-set enrichment analysis (GSEA), with GSEA software version 4.1.0 [19,20] and a gene set available from the Molecular Signatures Database (MolSigDB version 6.1). For GSEA analyses, the normalized enrichment score (NES) indicates the extent to which a gene set is overrepresented at the top or bottom of the ranked list of genes, normalized against the enrichment score for all permutations of the dataset. The false discovery rate (FDR) *q*-value and the family-wise error rate (FWER) *p*-value estimate the probability of the NES being a false-positive finding. The *p*-value lies between 0 and 1, with a value of 0.0 indicating that the *p*-value is less than 1/number-of-permutations. The FDR *q*-value corresponds to the *p*-value of the FDR adjusted for distribution of *p*-values. The FWER is a more stringent test than FDR, because it limits the probability of there being at least one false discovery.

## 3. Statistical Analysis

Flow cytometry data were analyzed in Wilcoxon or Mann–Whitney tests performed with JMP 14 software (SAS Institute Inc., Cary, NC, USA).

## 4. Results

### 4.1. CD4^+^CD57^+^PD1^−^ Cells Are Resistant to Belatacept

CD4^+^CD57^+^PD1^−^ T cells have been shown to be associated with a higher risk of ACR development in belatacept-treated patients. We therefore assessed the phenotype of these cells from patients awaiting transplantation. Both CD4^+^CD57^+^PD1^−^ and CD4^+^CD57^+^PD1^+^ T cells have a mostly memory phenotype (Figure 1A). These cell populations consist mostly of effector memory T cells, with only a small percentage of naive cells, much lower than for CD4^+^CD57^−^ T cells. We assessed the activation of these cells with healthy donor T lymphocytes separated on the basis of CD57 and then PD1 expression, stained with VPD450 and then cultured with activated aDCs, with or without belatacept. The proliferation of CD4^+^CD57^+^PD1^−^ and CD4^+^CD57^+^PD1^+^ T cells in culture was not inhibited by belatacept, whereas belatacept decreased the proliferation of CD4^+^CD57^−^ T cells by two thirds (Figure 1B). Both CD4^+^CD57^+^PD1^−^ and CD4^+^CD57^+^PD1^+^ T cells were considered to be belatacept-resistant, and they were pooled together for the rest of the study. These cells also expressed several markers of activation, such as CD25 and CD38 (Figure 1C). They also had a cytotoxic phenotype, with granzyme B and perforin expression, and displayed degranulation in the presence of a target, as shown by CD107a expression at the cell surface, as previously demonstrated [8,10].

### 4.2. Pathways Differentially Regulated by Belatacept during Early Activation of CD4^+^CD57 Sub Populations

We assessed the mechanism of proliferation of these cells, by comparing transcriptomes of CD4^+^CD57^+^PD1^−^, CD4^+^CD57^+^PD1^+^ and CD4^+^CD57^−^ T cells. Cells were incubated with aDCs, with or without belatacept, for 6 h, for the detection of early events. The pathways displaying differential regulation by belatacept in the different populations of CD4^+^ T cells are presented in Figure 2A. Interestingly, GSEA analysis indicated that several pathways involved in activation/survival were strongly inhibited by belatacept in CD4^+^CD57^−^ T cells but not in the CD4^+^CD57^+^PD1^−^ or CD4^+^CD57^+^PD1^+^ cells (Figure 2A). Three of them involved cell survival and activation gene sets and were of particular interest. These gene sets were MYC targets V1, mTORC1 pathway and IFN alpha response. Transcripts modulated in these gene sets by belatacept in CD4^+^CD57^−^ T cells are represented as networks (Figure 2B). Because mTOR is a master protein complex regulating T-cell proliferation, we focused on the mTOR pathway since it was dramatically affected in CD4^+^CD57^−^ when treated by belatacept but not in the CD4^+^CD57^+^ subsets. Likewise gene-set enrichment in untreated versus belatacept-treated cells was significantly observed in CD4^+^CD57^−^ cells but not in others cells, as shown by FDR-*q* values and FWER-*p* values (Figure 2C). The core genes affected by belatacept during CD4^+^CD57^−^ allogeneic activation were associated with activation and cell-cycle engagement in T cells (proteasome subunit) and metabolism (Figure 2D), as expected from the antiproliferative effect shown in Figure 1. In contrast to belatacept-treated CD4^+^CD57^−^ cells, in which few gene sets were increased, untreated CD4^+^CD57^−^ cells showed increased in cytokine-response, inflammation and cell-cycle related gene sets (Figure 2A). These results imply that mTORC1 signaling is not affected by belatacept in activated CD4^+^CD57^+^ cells, suggesting that this signaling activity may be involved in the belatacept resistance of these cells (Figure 2).

### 4.3. mTOR Pathway Is Activated in Belatacept-Resistant T Cells

To assess the expression of mTOR in the different subsets of CD4+ cells, mTOR expression was determined by flow cytometry. mTOR was slightly increased in CD4^+^CD57^+^ and CD8^+^ CD57^+^ positive cells as compared to CD4^+^CD57^−^ and CD8^+^CD57^−^ cells, respectively (*p* < 0.04) (Figure 3A). In addition, the activation of the mTORC1 pathway was evaluated through the determination of the phosphorylation of S6 ribosomal protein which is activated downstream mTORC1. At baseline, activation of S6 ribosomal protein in CD4^+^CD57^+^ T cells was slightly, but not significantly, higher than in CD4^+^CD57^−^ cells (*p* < 0.08) (Figure 3B). The basal level of CD8 activation was similar in CD8 subsets (Figure 3B). When activated through CD3 engagement, all CD4 and CD8 subsets exhibited a higher expression of phospho-S6. However, a significantly higher upregulation of phospho-S6 was observed in CD4^+^CD57^+^ cells as compared to CD4^+^CD57^−^ cells highlighting the implication of this pathway in CD4^+^CD57^+^ cells.

### 4.4. mTOR Inhibition Impairs T-Cell Proliferation and Activation of Both CD57^−^ and CD57^+^ T Cells

We then analyzed the effects of using an mTORi, everolimus, to inhibit mTOR. The concentration used was similar to that measured in treated transplant recipients (5 ng/mL). We found that the mTORi inhibited the proliferation of CD4^+^CD57^−^ T cells by 51%, and that of CD8^+^CD57^−^ T cells by 32%, but that it inhibited the proliferation of CD4^+^CD57^+^ T cells and CD8^+^CD57^+^ T cells less strongly, by 30% and 8%, respectively. The results obtained with a combination of belatacept and everolimus were similar to those obtained with everolimus alone (Figure 4A). Everolimus decreased the activation of CD4^+^ and CD8^+^ cells, as shown by assessments of the expression of CD38 (Figure 4B). Everolimus was more effective in CD4^+^CD57^+^ and CD8^+^CD57^+^ cells (56 and 24% decreases, respectively). The addition of belatacept did not decrease the activation of the CD4^+^CD57^+^ population and only slightly decreased activation of the CD8^+^CD57^+^ population.

CD4^+^CD57^+^ cells have been shown to be cytotoxic and to express granzyme B (GrzB). As previously described, more than 50% of CD4^+^CD57^+^ T cells and more than 45% of CD8^+^CD57^+^ T cells expressed GrzB, whereas the percentage of cells expressing GrzB was below 12% for CD4^+^CD57^−^ T cells and 20% for CD8^+^CD57^−^ T cells. Everolimus decreased GrzB expression more strongly (by 37%) in CD4^+^CD57^+^ T cells (*p* = 0.0039) than in CD8^+^CD57^+^ T cells (27%), but even this second decrease was significant (Figure 5A; *p* = 0.0078). We assessed the cytotoxic functions of these cells by incubating them with everolimus for five days in the presence of activated allogeneic DCs and placing them in the presence of target cells from an aDC donor. Cytotoxicity was assessed by evaluating the membrane expression of CD107a, a marker of degranulation (Figure 5B). In this context, a high proportion of CD4^+^CD57^+^ T cells presented CD107a staining (45%), whereas staining was much weaker for the total CD4 population. The rate of CD107a-positive cells was considerably decreased by prior treatment with everolimus (*p* = 0.0039). Similarly, the proportion of CD8^+^CD57^+^ cells displaying degranulation was higher than that for the total CD8^+^ population. However, despite a significant decrease in the proportion of cells displaying CD107a staining in the presence of everolimus, this decrease appeared to be smaller than for CD4^+^CD57^+^ cells (27% vs. 52%).

### 4.5. mTOR Inhibition Regulates T-Cell Activation in Patients Treated with Belatacept

Three patients treated with belatacept (two lung transplant patients and one kidney transplant patient) who presented ACR received an mTOR inhibitor (mTORi) to replace mycophenolate mofetil (MMF). Their characteristics are shown in Table 1. All three patients were rapidly switched onto belatacept shortly after transplantation, due to thrombotic microangiopathy related to calcineurin inhibitor use, as their complement fractions, I and H factors and ADAMST 13 proteins were in the normal range. All three developed acute rejection following this switch, with a recurrence of rejection in one patient. The antimetabolite treatments of these patients were replaced with mTORi, in combination with belatacept and steroids. Following the introduction of mTORi in association with belatacept, none of these patients displayed a recurrence of ACR or a relapse of thrombotic microangiopathy. All three patients had a peripheral lymphocyte phenotype with a high percentage of activated CD4 and CD8 T cells expressing CD38 and CD57 at the time of mTORi introduction. The peripheral T-cell phenotypes of these patients are reported in Table 2. Most of the T lymphocytes of these patients expressed CD38, a marker of activation, at the cell surface at the time of ACR. Following mTORi introduction, the proportion of CD4^+^CD57^+^ cells did not decrease significantly. By contrast, the proportion of CD4^+^ T cells that were CD38^+^ decreased significantly, as did the proportion of CD8^+^CD38^+^ T cells (Figure 6). This decrease in T-cell activation following mTORi treatment in vivo should be considered in light of the inhibition of the different T-cell subsets in vitro.

## 5. Discussion

Belatacept is a checkpoint inhibitor that has been shown to preserve renal function and to improve patient and graft survival in renal transplant patients [2,3,5,21]. However, it is associated with a higher risk of ACR, which has limited its use [7,8,22]. The higher acute rejection rates in belatacept-treated patients have been associated with the presence, before renal transplantation, of T cells with a memory phenotype corresponding to CD4 effector memory cells [7,8,23]. We also found that the cells associated with ACR expressed the CD57 marker. We show here that CD57^+^ T cells are resistant to belatacept in an in vitro mixed lymphocyte reaction model with activated allogeneic aDCs [8,10]. After a short period of activation (6 h), for the detection of early regulatory signals occurring in activated belatacept-sensitive but not belatacept-resistant T cells, we observed a downregulation of the mTOR pathway in belatacept-sensitive T cells. The mTOR pathway plays an important role in the control of T-cell expansion [15]. Ahmed and coworkers showed, in an in vivo model, that these cells were not exhausted and were able to proliferate [24]. Activation of T cells is dependent on several signals, some specific to antigens and others dependent on co-stimulation, pro-inflammatory cytokines or TLRs [25,26,27,28]. Most of these signals are converging to mTOR complex activation, which controls growth, nutrient uptake required for T-cell activation, differentiation and expansion. Thus, several pathways of T-cell activation, whose importance depends on T-cell subsets, imply mTOR activation. We observed that belatacept, which impairs the CD28 co-signal pathway, reduced the mTOR signaling pathway in CD4^+^CD57^−^ T cells but not in belatacept-resistant T cells. Moreover, CD4^+^CD57^+^ T cells have a basal activation of mTOR that can be related to cytokine signaling, as recently reported [10]. Our results indicate that this pathway is impaired in belatacept-sensitive T cells but not in belatacept-resistant T cells. Treatments controlling the mTOR pathway in belatacept-resistant T cells would therefore be of potential interest for the efficient inhibition of these T cells. In vitro, we evaluated mTOR inhibition, using a concentration of 5 ng/mL everolimus, which corresponds to the trough level in patients (3–8 ng/mL) and is at the lower end of the concentrations used in published in vitro studies (5 to 100 nM). However, a closer analysis of the dose-dependence of the response of each T-cell subpopulation would improve our understanding of the impact of everolimus on T-cell activation and cytotoxic properties, as not all the everolimus circulating in the blood is available due to interaction with proteins and other cell types. Surprisingly, we found that an mTOR inhibitor impaired the activation and cytotoxicity of belatacept-resistant T cells in vitro and in vivo, opening up possibilities for new therapeutic options for belatacept-based regimens.

Our results are consistent with those of Castro-Rojas and coworkers, who showed that mTORi were able to control acute rejection in patients treated with belatacept [29]. Interestingly, they also found that their patients had a high frequency of CD8^+^ T_EM_ cells expressing high levels of CD38, a marker of activation. They found that mTORi treatment was associated with a decrease in CD38 expression by T cells. We also observed a decrease in CD38 expression in both CD4^+^ and CD8^+^ T cells, not only in vitro, but also in vivo. We also found that mTORi treatment was associated with a decrease in the expression of granzyme B and with a decrease in the cytotoxic function of CD4^+^ T cells, providing additional support for the idea that mTORi could be used, in association with belatacept, to control the alloreactive response. Conversely, the decrease in T-cell proliferation, both in vitro and in vivo, was significant, but smaller than that in the absence of mTORi, suggesting that the major effect of mTORi in association with belatacept is the regulation of T-cell activation. This finding is consistent with previous results showing that the mTORi currently available are active principally against the mTORC1 pathway, impairing T-cell metabolism and function, whereas the mTORC2 pathway regulates cell-cycle entry more tightly [30,31]. New molecules inhibiting both pathways, such as CC214-1 or CC214-2, are therefore of potential interest, as they might yield a stronger effect [12]. Thus, mTORi may be a more appropriate option than MMF for the treatment of refractory ACR in patients treated with belatacept. As shown both by Castro-Rojas et al. and here, mTORi not only impair the clonal expansion of alloreactive T cells, but they also regulate the activation and differentiation of T cells [29].

Interestingly, mTORi have been proposed for maintenance therapy in patients treated with belatacept after induction with anti-thymoglobulin antibodies. Two pilot studies reported low rates of acute rejection in the groups of patients treated with belatacept and mTORi, similar to those for standard care based on a combination of calcineurin inhibitors and MMF, but significantly lower than those in patients treated with belatacept plus MMF. Consistent with our results, this suggests complementarity between the elements of this combined treatment, providing better coverage of the T-cell repertoire, including control over belatacept-refractory T cells, in patients [16,18,32]. Additional larger, randomized studies would be required to confirm this hypothesis.

The results of this study reveal an important role of mTORi in the control of CD4^+^ and CD8^+^ T-cell activation, particularly for cells resistant to belatacept. Similar findings were obtained for a small number of patients for whom calcineurin inhibitors were contraindicated due to calcineurin-inhibitor-associated thrombotic microangiopathy. These promising results require confirmation in a larger number of patients, but they open up new opportunities for patients with a contraindication for calcineurin inhibitor use and in patients with acute rejection, for whom therapeutic options are limited.

## Figures and Tables

**Figure 1 pharmaceutics-15-01299-f001:**
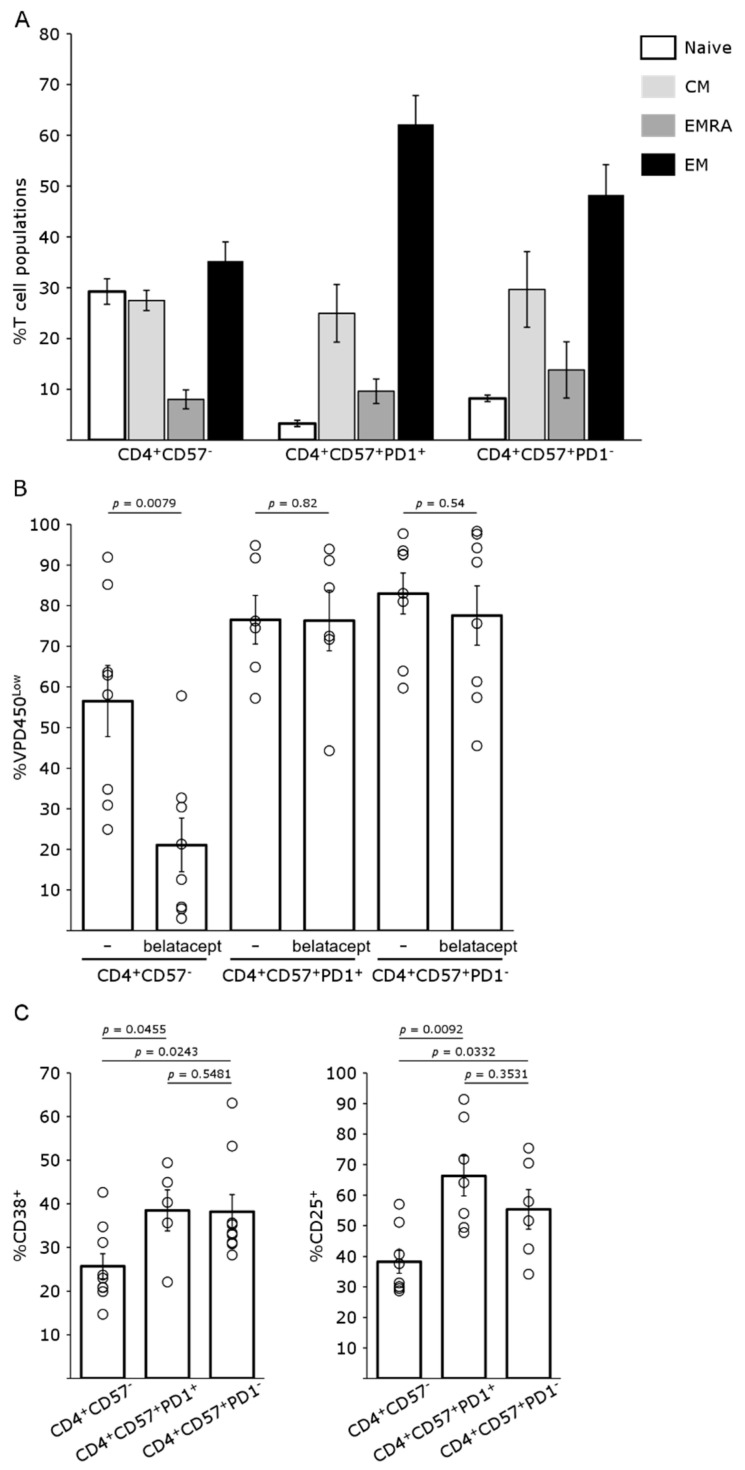
CD4^+^ T-cell phenotypes, proliferation and activation. (**A**) The CD57 and PD1 expression phenotypes of T cells collected from renal transplant recipients before transplantation were assessed by flow cytometry. Naïve (CCR7^+^CD45RO^−^), central memory (CM; CCR7^+^CD45RO^+^), effector memory re-expressing CD45RA (EMRA; CCR7^−^CD45RO^−^), and effector memory cells (EM; CCR7^−^CD45RO^+^) were characterized according to their expression of CD45R0 and CCR7. Mean (bar chart) and SEM of six independent experiments. (**B**) Proliferation of CD4^+^ T-cell populations (VPD450 dilution) from healthy donor T cells assessed by flow cytometry after 5 days of MLR in the presence and absence of belatacept. Data for six (CD4^+^CD57^+^PD1^+^) and eight (CD4^+^CD57^−^ and CD4^+^CD57^+^PD1^−^) independent experiments (pictograms) are shown, with the mean (bar chart) and SEM. (**C**) The expression of CD38 and CD25, as markers of activation, in the various CD4^+^ T-cell populations from healthy donors were assessed, after 5 days of MLR, by flow cytometry in six independent experiments (pictograms). The mean (bar chart) and SEM are shown.

**Figure 2 pharmaceutics-15-01299-f002:**
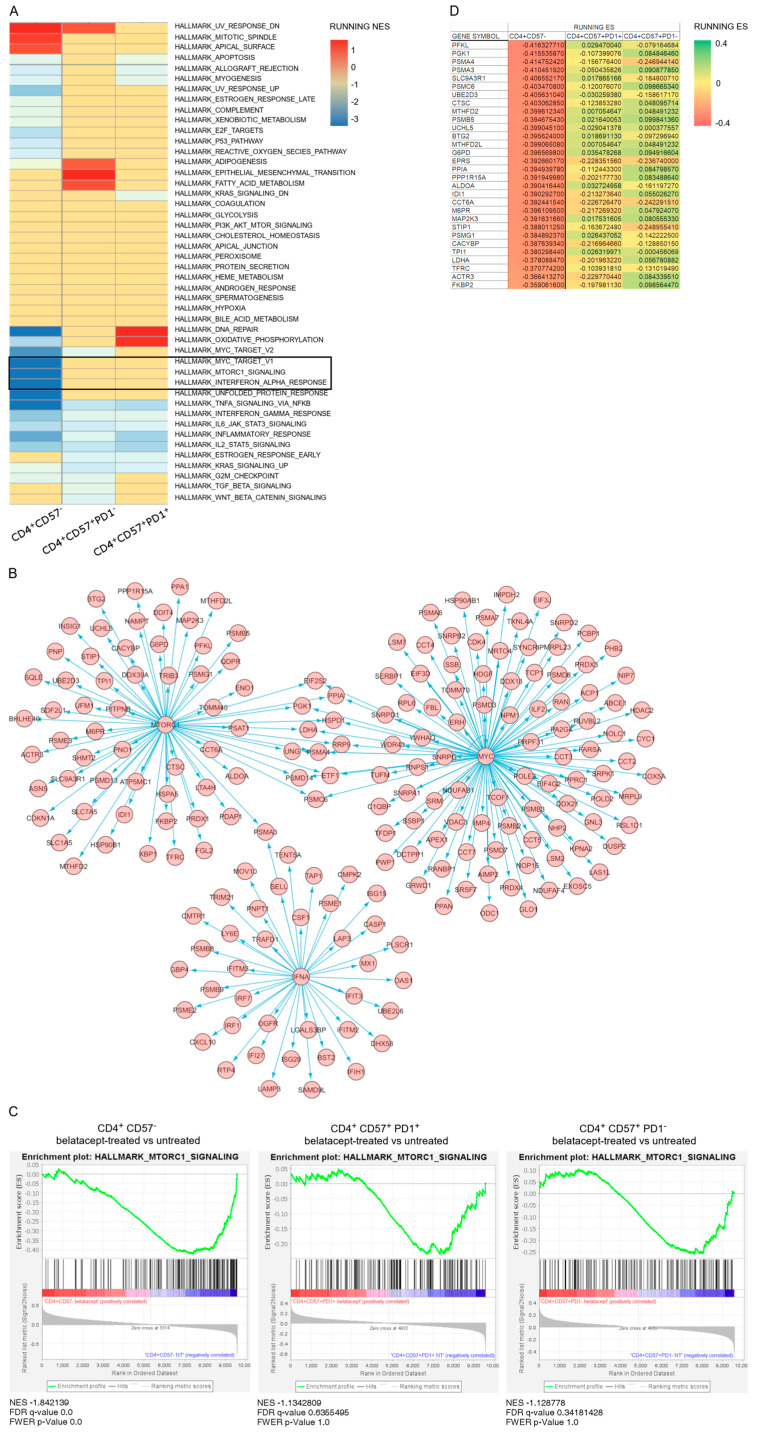
Transcriptomic analysis of the effect of belatacept on CD4^+^CD57^−^, CD4^+^CD57^+^PD1^−^ and CD4^+^CD57^+^PD1^+^ allogeneic T cells after 6 h of culture with activated dendritic cells. (**A**) Heatmap of normalized enrichment score (NES) of untreated versus belatacept-treated CD4 subpopulations analyzed by gene-set enrichment analysis (GSEA). (**B**) Network representation of gene sets MYC target V1, mTORC1 signaling and IFN alpha response profiles of untreated CD4^+^CD57^−^ versus belatacept-treated. (**C**) mTORC1 signaling enrichment plot for CD4^+^CD57^−^, CD4^+^CD57^+^PD1^+^ and CD4^+^CD57^+^PD1^−^cells activated in the presence and absence of belatacept. Normalized enrichment score (NES), false discovery rate (FDR-*q* value) and family-wise error rate (FWER *p*-value) of the hallmark mTORC1 signaling gene set for the various CD4^+^ populations, in the presence and absence of belatacept. Belatacept significantly impaired the transcription of mTORC1 pathway genes only in CD4^+^CD57^−^ cells. (**D**) Heatmap of the first 30 genes of the leading-edge subset of the mTORC1 signaling enrichment plot ranked for CD4^+^CD57^−^ cells untreated versus belatacept-treated. The leading-edge subset is the core of the gene set accounting for the enrichment signal. Enrichment score at the position in the ranked list of genes (RUNNING ES) of CD4^+^CD57^+^PD1^+^ and CD4^+^CD57^+^PD1^−^ have been ordered as a CD4^+^CD57^−^ list and plotted on the heat map allowing comparisons.

**Figure 3 pharmaceutics-15-01299-f003:**
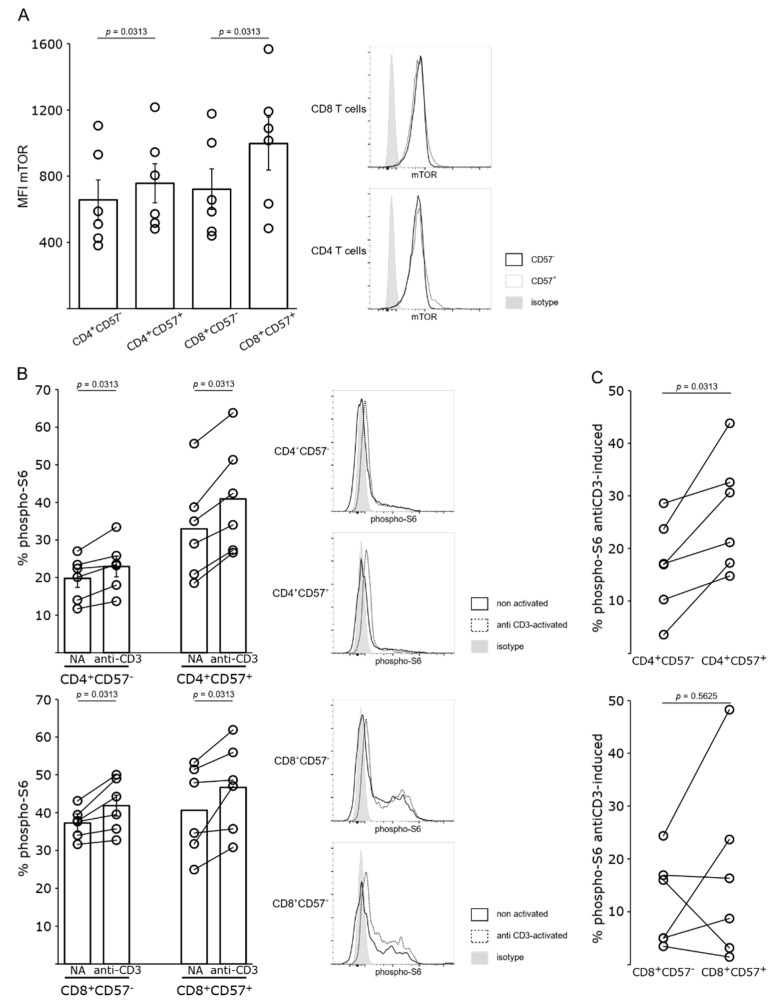
mTOR expression and activation of mTORC1 pathway in CD57^+^ and CD57^−^ T cells. (**A**) mTOR expression by CD4^+^CD57^−^, CD4^+^CD57^+^, CD8^+^CD57^−^ and CD8^+^CD57^+^ T cells was analyzed by flow cytometry. Left panel, mean of fluorescence intensity (MFI) of cells of six healthy donors, the mean (bar chart) and SEM are shown. Right panel, representative flow cytometry profile. (**B**) Phosphorylation of S6 in CD4^+^CD57^−^, CD4^+^CD57^+^, CD8^+^CD57^−^ and CD8^+^CD57^+^ T cells were analyzed with and without anti-CD3 activation by flow cytometry. Left panel, percentage of positive cells of six healthy donors, the mean (bar chart) and SEM are shown. Right panel, representative flow cytometry profile. (**C**) Percentage of S6-phosphorylation induced by anti-CD3 activation calculated for each experiment was depicted.

**Figure 4 pharmaceutics-15-01299-f004:**
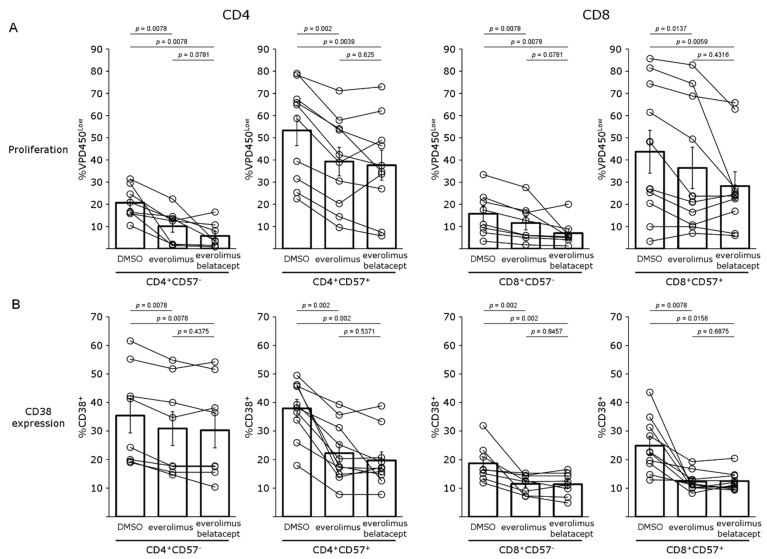
Activation and proliferation of CD4^+^CD57^−^, CD4^+^CD57^+^, CD8^+^CD57^−^ and CD8^+^CD57^+^ T cells in the presence of mTOR inhibitor: (**A**) Proliferation and (**B**) CD38 expression of CD4^+^CD57^−^, CD4^+^CD57^+^, CD8^+^CD57^−^ and CD8^+^CD57^+^ T cells after 5 days of culture with allogeneic aDCs without (DMSO) or with everolimus, or everolimus + belatacept. Eight (CD57−) or ten (CD57+) independent experiments (pictograms) were performed, and the mean (bar chart) and SEM are shown.

**Figure 5 pharmaceutics-15-01299-f005:**
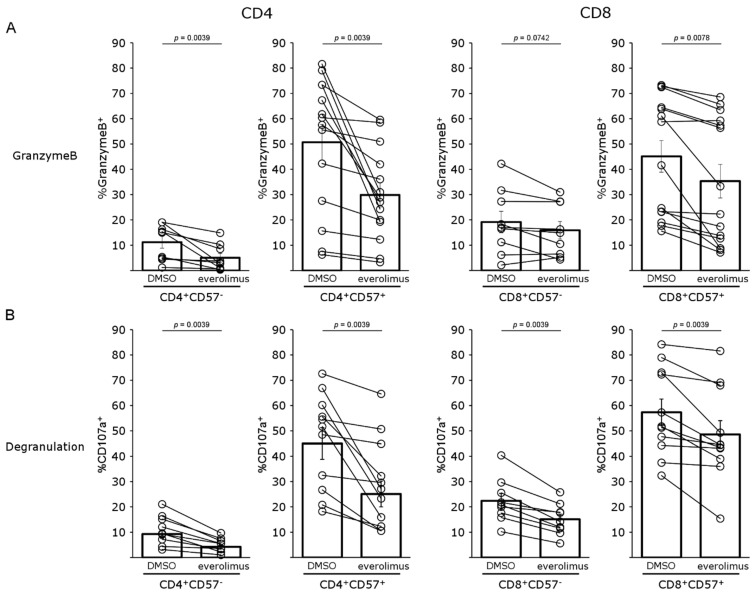
Granzyme B expression and cytotoxicity in CD4^+^CD57^−^, CD4^+^CD57^+^, CD8^+^CD57^−^ and CD8^+^CD57^+^ T cells treated with mTOR inhibitor: (**A**) Expression of granzyme B and (**B**) CD107a by CD4^+^CD57^−^, CD4^+^CD57^+^, CD8^+^CD57^−^ and CD8^+^CD57^+^ T cells after 5 days of culture with allogeneic aDCs without (DMSO) or with everolimus, followed by incubation with target cells for 4 h. Nine (CD57^−^) and 14 (CD57^+^) (**A**) or 9 (CD57^−^) and 11 (CD57+) (**B**) independent experiments (pictograms) were performed. The mean (bar chart) and SEM are shown.

**Figure 6 pharmaceutics-15-01299-f006:**
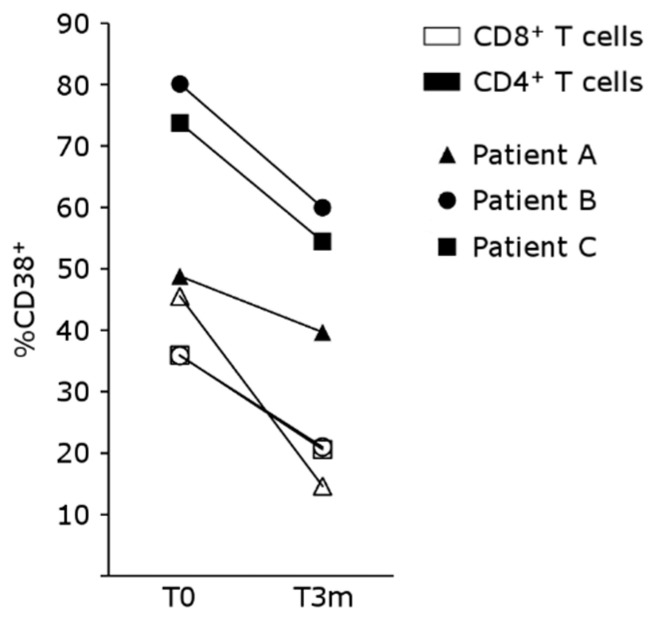
CD38 expression on CD4^+^ or CD8^+^ T cells from patients treated with belatacept, before (T0) or 3 months after (T3m) the introduction of everolimus (*n* = 3 patients).

**Table 1 pharmaceutics-15-01299-t001:** Summary table of patient characteristics and treatments.

	A	B	C
Age	59	43	71
Sex	M	F	M
Disease	Idiopathic pulmonary fibrosis	Sarcoidosis with pulmonary fibrosis	Nephroangiosclerosis
Type of Tx	Lung	Lung	Kidney
Age at Tx (years)	58	43	70
Transplantation rank	1	2	1
Induction therapy	rIL2	Thymoglobulin	Thymoglobulin
Maintenance therapy			
CNI	Cyclosporin	Cyclosporin	Tacrolimus
Anti-metabolites	MMF	MMF	MMF
Steroids	Yes	Yes	Yes
Rejection Y/N	N	N	N
Time to the switch to belatacept (months)	2	2	3
Reason for switch	TMA	TMA	TMA
eGFR at switch	19	55	22
eGFR after switch (3 months)	59	84	34
Maintenance therapy after switch to			
belatacept	Yes	Yes	Yes
Anti-metabolites	MMF	Azathioprine	MMF
Steroids	Yes	Yes	Yes
Other treatments	PEx—soliris	PEx—oliris	PEx
DSA at switch	Neg	Pos (MFI 2000)	Neg
De novo DSA after switch	Neg	Neg	Neg
Number of rejection episodes since switch to belatacept	1	2	1
Treatment of rejection	Steroids 1 mg/kg, with a stepwise decrease	For the first episode, an increase in steroid dose. For the second episode, none, because of concomitant infection	Steroid boluses
Time to imTOR introduction (months)	16	14	12
Certican trough concentration (ng/mL)	3.7	2	4.9

Tx: transplantation; CNI: calcineurin inhibitor; TMA: thrombotic microangiopathy; MMF: mycophenolate mofetil; PEx: plasma exchange; MFI: mean fluorescence intensity.

**Table 2 pharmaceutics-15-01299-t002:** Phenotype of peripheral lymphocytes assessed before and 3 months after mTORi introduction.

	A		B		C	
	T0	T3m	T0	T3m	T0	T3m
CD3 in Ly45 (%)	87.6	86.7	79.7	75.4	88.2	88.1
CD4 in CD3 (%)	23.4	15.3	20.6	35.3	28.1	34.7
CD45RA^−^CCR7^+^ (CM) in CD4 (%)	32.7	37.2	57.7	57.6	28.2	40.4
CD45RA^+^CCR7^+^ (naive) in CD4 (%)	41.7	42.5	15.2	21	2.9	4.3
CD45RA^−^CCR7^−^ (EM) in CD4 (%)	23.2	17.1	26.9	20	50.2	40.6
CD45RA^+^CCR7^−^ (EMRA) in CD4 (%)	2.1	3	0.1	0.6	18.4	14.5
HLA-DR in CD4 (%)	0.7	2	9.4	9	34.4	16.5
CD57 in CD4 (%)	3.8	10.7	9.3	9.7	39.4	29.8
CD38 in CD4 (%)	45.5	14.6	35.9	20.6	35.8	20.9
CD8 in CD3 (%)	72.9	80.9	52.4	44.6	65.5	59.4
CD45RA^−^CCR7^+^ (CM) in CD8 (%)	3.7	1.3	1.8	3.5	1.5	2.8
CD45RA^+^CCR7^+^ (naive) in CD8 (%)	7.5	5.6	7.2	16.7	0.8	1.2
CD45RA^−^CCR7^−^ (EM) in CD8 (%)	41.3	27.2	53.5	27.7	18.8	30.3
CD45RA^+^CCR7^−^ (EMRA) in CD8 (%)	47.2	65.8	37.4	52	78.7	65.5
HLA-DR in CD4 (%)	6.2	15.1	42	7	64	38.9
CD57 in CD8 (%)	71.1	82.7	43.4	49.5	63.7	54
CD38 in CD8 (%)	48.8	39.6	73.7	54.4	80.1	59.9

## Data Availability

The data presented in this study are available on request from the corresponding author. The data are not publicly available due to privacy.

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
