# Peer review of "mTOR Inhibition Impairs the Activation and Function of Belatacept-Resistant CD4+CD57+ T Cells In Vivo and In Vitro"

_pharmaceutics, 2023, doi:10.3390/pharmaceutics15041299_

Round 1

Reviewer 1 Report

The authors demonstrate that the combination of mTOR inhibitor with belatacept can be advantageous in patients that develop ACR by inhibiting belatacept-resistant T cells. Using in vitro studies, they show that mTOR pathway is activated in belatacept-resistant T cells. They identify CD57+ T cells as resistant to belatacept. They present human patient data to solidify their findings. Although the human data presented in this article is only from three patients, it puts forward a convincing case of using mTOR inhibitors in combination with belatacept, specifically in case of ACR. Overall, the article follows scientific logic and should be of interest to the readership of Therapeutics. The article can be considered for publication after addressing the following minor issues:

1.       Throughout the results section the titles need to be improved, some of the subsections have titles of one to three words, which is not sufficient. The title needs to be self-explanatory.

2.       In Fig 1, including the dot plots from flow cytometry as an example of gating strategy will be useful addition.

3.       In Fig2, authors show enrichment plots for mTOR signaling pathway, how was this pathway selected based on the sequencing data. Were there other pathways that were identified from the GSEA analysis. Including a heatmap of the leading-edge gene subset comparing their expression between CD4+CD57- and CD4+CD57+ T cells will be more useful.

Author Response

Reviewer 1 :

The authors demonstrate that the combination of mTOR inhibitor with belatacept can be advantageous in patients that develop ACR by inhibiting belatacept-resistant T cells. Using in vitro studies, they show that mTOR pathway is activated in belatacept-resistant T cells. They identify CD57+ T cells as resistant to belatacept. They present human patient data to solidify their findings. Although the human data presented in this article is only from three patients, it puts forward a convincing case of using mTOR inhibitors in combination with belatacept, specifically in case of ACR. Overall, the article follows scientific logic and should be of interest to the readership of Therapeutics. The article can be considered for publication after addressing the following minor issues:

  1. Throughout the results section the titles need to be improved, some of the subsections have titles of one to three words, which is not sufficient. The title needs to be self-explanatory.

R: We have added titles to the main paragraphs to improve the understanding and the presentation as suggested.

  1. In Fig 1, including the dot plots from flow cytometry as an example of gating strategy will be useful addition.

R: We agree and we have added a gating strategy as a supplementary figure.

  1. In Fig2, authors show enrichment plots for mTOR signaling pathway, how was this pathway selected based on the sequencing data. Were there other pathways that were identified from the GSEA analysis. Including a heatmap of the leading-edge gene subset comparing their expression between CD4+CD57- and CD4+CD57+ T cells will be more useful.

R: As highlighted by the reviewer, few pathways were differentially regulated. Following the GSEA analysis, several pathways were strongly inhibited by belatacept in CD4+CD57- T cells but not in the CD4+CD57+Pd1- or CD4+CD57+Pd1+. We have now provided a heatmap comparing these 3 populations (Figure 2A). Three major pathways of activation/survival were corresponding to this situation: MYC targets V1, mTOR pathway, IFN alpha pathway. Unfolded protein pathway was not directly related to cell activation and was avoided. Molecules that were affected in these cells are represented as networks (Figure 2B). Since mTOR pathway is an important pathway that controls T cell proliferation and for which available inhibitory molecules could be used, we decided to evaluate the implication of this pathway. In addition, the involvement of the mTOR pathway was confirmed in these cells by flow cytometry. mTOR was slightly more expressed in CD4+CD57+ and CD8+/CD57+ than in CD4+CD57- and CD8+CD57- T cells. Moreover, the activated downstream S6 protein (phospho-S6 (pS6)) tend to be higher in the in the CD4+CD57+ than in the CD4+CD57- T cells. In addition, following an activation through CD3, phosho-S6 protein is upregulated in both populations of cells but was superior inCD4+CD57+ T cells (Figure 3). In CD8 Tcells, the percentage of mTOR and phospho-mTOR were similarly expressed in both cell types.

Reviewer 2 Report

In the manuscript entitled “mTOR inhibition impairs the activation and function of belatacept-resistant CD4+CD57+ T cells in vivo and in vitrothe authors investigate the mechanisms behind acute cellular rejection from the immunosuppressive drug Belatacept by performing transcriptomic analysis of in-vitro-activated cells in belatacept-sensitive cells versus belatacept-resistant T cells. They discover that mTOR is downregulated in belatacept-sensitive cells and therefore propose that mTOR inhibitor could be used in association with belatacept, to prevent acute cellular rejection in cases of calcineurin intolerance. Overall, the manuscript is well-written and presents interesting results that support the conclusions. As the authors acknowledge, one weakness is the small number of patients, however, I think this is acceptable since they support the mechanistic conclusions of the in vitro studies. The main premise of the manuscript has been reported before (i.e. see PMID: 25959589), thus the authors need to stress what is novel in their study more clearly. Is it the transcriptomic analysis? If yes, then what other insights does this analysis provide apart from mTOR?

Additional comments:

Figure 2, is at low resolution and difficult to see.

While the transcriptomic analysis is important, it may not accurately reflect the levels of the proteins that participate in the biochemical pathways studied. Can it be supplemented by proteomic analysis or at least check the levels of proteins suggested to be affected more directly (i.e. Western Blot?)

Page 14 “The results obtained with a combination of belatacept and everolimus were similar to those obtained with everolimus alone” – this does not seem to be the case: the effect appears additive – please elaborate

Targeting mTOR with everolimus is not the same as downregulating mTOR as evidenced by transcriptome analysis. Protein levels need to be checked in both cases.

The paper could use additional mechanistic insight. Why is the mTOR pathway inhibited in some T cells? How does mTOR inhibition lead to sensitization versus belatacept?

Author Response

Reviewer 2

In the manuscript entitled “mTOR inhibition impairs the activation and function of belatacept-resistant CD4+CD57+ T cells in vivo and in vitro” the authors investigate the mechanisms behind acute cellular rejection from the immunosuppressive drug Belatacept by performing transcriptomic analysis of in-vitro-activated cells in belatacept-sensitive cells versus belatacept-resistant T cells. They discover that mTOR is downregulated in belatacept-sensitive cells and therefore propose that mTOR inhibitor could be used in association with belatacept, to prevent acute cellular rejection in cases of calcineurin intolerance. Overall, the manuscript is well-written and presents interesting results that support the conclusions. As the authors acknowledge, one weakness is the small number of patients, however, I think this is acceptable since they support the mechanistic findings of the in vitro studies. The main premise of the manuscript has been reported before (i.e. see PMID: 25959589), thus the authors need to stress what is novel in their study more clearly. Is it the transcriptomic analysis? If yes, then what other insights does this analysis provide apart from mTOR?

R: In this article, we have focused our research on pathways that were affected by belatacept in CD4+CD57- T cells but not in the other cells (CD4+CD57+Pd1- or CD4+CD57+Pd1+ T cells) suggesting a continuous involvement of these corresponding pathways in CD4+CD57+Pd1- or CD4+CD57+Pd1+ T cells. As suggested by both reviewers, we are showing a heatmap showing the affected and non-affected pathways in the different populations of cells (Figure 2A) which highlights pathways of interest. Three major pathways of activation/survival were corresponding to this situation: MYC targets V1, mTOR pathway, IFN alpha pathway. Unfolded protein pathway was not directly related to cell activation and was avoided. Since mTOR pathway is an important pathway for controlling T cell proliferation and for which available inhibitory molecules could be used, we decided to evaluate the implication of this pathway. Now, we also showed that mTOR is sightly more expressed in CD4+CD57+ cells than in CD4+CD57- cells and that mTOR pathway is activated (phosphorylation of S6kinase) at baseline and after a short activation in CD4+CD57+ T cells. After activation, pS6 kinase is more importantly expressed in CD4+CD57+ cells as compared to CD4+CD57- cells (Figure 3)

Additional comments:

Figure 2, is at low resolution and difficult to see.

R: We agree. To improve the reading, we have increased the size of GSEA figure. However GSEA generated figures with low definition when exported on html.

While the transcriptomic analysis is important, it may not accurately reflect the levels of the proteins that participate in the biochemical pathways studied. Can it be supplemented by proteomic analysis or at least check the levels of proteins suggested to be affected more directly (i.e. Western Blot?)

R: We agree that the transcriptomic analysis is not accurately reflecting the level of proteins, but pathways that are affected. Because most of the initial modifications are in relation to cell signaling, we have analyzed the expression of mTOR and the activation of the mTOR pathway. We show that S6 kinase, which acts downstream mTOR-C1, is more importantly phosphorylated in CD4+CD57+ T cells than in CD4+CD57- T cells with or without CD3 activation (Figure 3). This indicated the involvement of the mTOR-C1 activation in CD4+CD57+ T cells.

Page 14 “The results obtained with a combination of belatacept and everolimus were similar to those obtained with everolimus alone” – this does not seem to be the case: the effect appears additive – please elaborate

R: We agree that the effect of Everolimus + Belatacept vs Everolimus alone is not exactly the same for CD4 or CD8 T cells. We thank the reviewer for it remark. We have modified our analysis of the results: The effect of the combination of belatacept and everolimus was similar to that obtained with everolimus alone for CD57+ CD4+ T cells but tend to be higher for CD57+ CD8+ T cells.

Targeting mTOR with everolimus is not the same as downregulating mTOR as evidenced by transcriptome analysis. Protein levels need to be checked in both cases.

R: As suggested, we have analyzed the level of expression of mTOR in CD4+CD57- vs CD4+CD57+ population. At baseline, we observed a similar level of mTOR expression in these 2 populations, but we observed that the mTOR-C1 pathway (p-S6 kinase) was more constitutively activated in CD4+CD57+ T cells than in CD4+CD57- T cells.

The paper could use additional mechanistic insight. Why is the mTOR pathway inhibited in some T cells? How does mTOR inhibition lead to sensitization versus belatacept?

R: We have introduced a paragraph in the discussion to highlight different pathways of activation of mTOR in the different subsets of T cells. Activation of T cells is depending on several signals, some of them being specific of antigens and others being dependent on (2) co-stimulation, (3) pro-inflammatory cytokines or TLRs. Most of them are converging to the mTOR activity, that control growth, nutrient uptake required for T-cell activation,  differentiation, and expansion. Thus, several pathways of T cell activation, whose importance are depending on T cell subsets imply mTOR activation. We observed that belatacept which impairs the CD 28 co-signal pathway reduces mTOR signaling pathway in CD4+CD57- T cells but not in resistant T cells. Moreover, CD4+CD57+ T cells have a basal activation of mTOR that can be related to cytokine signaling as recently reported (herr F. et al.).  Therefore, the inhibition of mTOR would be of interest to control the proliferation and or activation of both CD4 and CD8 T cells.

The differential effect of belatacept and mTOR inhibition on sensitization is an important question implying different pharmacology of these 2 molecules but also different target cells. On one hand, belatacept impairs specifically the activation of naïve T cells or B cells by blocking the second signal of activation and limiting T cells activation and production of antibodies by B cells. On the other hand, mTOR inhibitors are required for the proliferation of lymphocytes but it less efficient to control B cell activation.
